# Wintertime Simulations Induce Changes in the Structure, Diversity and Function of Antarctic Sea Ice-Associated Microbial Communities

**DOI:** 10.3390/microorganisms10030623

**Published:** 2022-03-15

**Authors:** Violetta La Cono, Francesco Smedile, Francesca Crisafi, Laura Marturano, Stepan V. Toshchakov, Gina La Spada, Ninh Khắc Bản, Michail M. Yakimov

**Affiliations:** 1Institute of Polar Sciences, CNR, 98122 Messina, Italy; violetta.lacono@cnr.it (V.L.C.); francesca.crisafi@cnr.it (F.C.); l.marturano93@gmail.com (L.M.); gina.laspada@cnr.it (G.L.S.); 2Kurchatov Genome Center, NRC “Kurchatov Institute”, 123098 Moscow, Russia; stepan.toshchakov@gmail.com; 3Institute of Marine Biochemistry, VAST, Nghia Do, Hanoi 10000, Vietnam; ninhkhacban@vast.vn

**Keywords:** Antarctica, sea-ice brine, microbial community, SIMCO, sulfate-reducing bacteria, sulfur-oxidizing bacteria, SSU gene amplicon sequencing

## Abstract

Antarctic sea-ice is exposed to a wide range of environmental conditions during its annual existence; however, there is very little information describing the change in sea-ice-associated microbial communities (SIMCOs) during the changing seasons. It is well known that during the solar seasons, SIMCOs play an important role in the polar carbon-cycle, by increasing the total photosynthetic primary production of the South Ocean and participating in the remineralization of phosphates and nitrogen. What remains poorly understood is the dynamic of SIMCO populations and their ecological contribution to carbon and nutrient cycling throughout the entire annual life of Antarctic sea-ice, especially in winter. Sea ice at this time of the year is an extreme environment, characterized by complete darkness (which stops photosynthesis), extremely low temperatures in its upper horizons (down to −45 °C) and high salinity (up to 150–250 psu) in its brine inclusions, where SIMCOs thrive. Without a permanent station, wintering expeditions in Antarctica are technically difficult; therefore, in this study, the process of autumn freezing was modelled under laboratory conditions, and the resulting ‘young ice’ was further incubated in cold and darkness for one month. The ice formation experiment was primarily designed to reproduce two critical conditions: (i) total darkness, causing the photosynthesis to cease, and (ii) the presence of a large amount of algae-derived organic matter. As expected, in the absence of photosynthesis, the activity of aerobic heterotrophs quickly created micro-oxic conditions, which caused the emergence of new players, namely facultative anaerobic and anaerobic microorganisms. Following this finding, we can state that Antarctic pack-ice and its surrounding ambient (under-ice seawater and platelet ice) are likely to be very dynamic and can quickly respond to environmental changes caused by the seasonal fluctuations. Given the size of Antarctic pack-ice, even in complete darkness and cessation of photosynthesis, its ecosystem appears to remain active, continuing to participate in global carbon-and-sulfur cycling under harsh conditions.

## 1. Introduction

Antarctic sea-ice can cover up to 20 million km^2^ of the Southern Oceans, which comprises approximately 13% of the Earth’s surface area [1]. It represents a complex ecosystem consisting of platelet- and pack-ice, together with a sophisticated channel system, filled with hypersaline brines up to 5–8 times saltier than the surrounding semi-solid platelet-ice–seawater matrix [2,3,4,5]. The salinity of the brine within the channels is inversely proportional to the ice temperature [6] and presents remarkable seasonal changes. For example, it is known that winter sea ice can reach temperatures as low as −45 °C in its upper horizons, with a corresponding salinity of up to 237‰ [7]. Thus, the uppermost section of the pack ice experiences the most drastic changes during winter, and the concentration of salts in the brine sometimes overcomes the saturation point of salt. At this point, a change in the chemical composition of brine is driven by the successive precipitation of different salts, such as ikaite (CaCO_3_ × 6H_2_O), mirabilite (Na_2_SO_4_ × 10H_2_O) and hydrohalite (NaCl × 2H_2_O) following a temperature decrement of −2.2 °C, −8.2 °C and −22.9 °C, respectively [6,8].

Antarctic sea-ice is a very biologically active and productive habitat [9,10]. It is inhabited by a complex assemblage of organisms, coined as sea-ice microbial communities (SIMCO), that mainly consists of ice-adapted diatom species and prokaryotic organisms; however, it can also include allochthonous and autochthonous protists, zooplankton and small metazoa (crustaceans, worms) [11,12,13,14,15]. SIMCOs have been shown to have a major role in trophic food webs, with a plethora of organisms at different development stages finding shelters and food within ice floes, and near the ice—seawater interface [15]. At the ecological level, SIMCOs can act as a reservoir of organisms, able to colonize the sea-ice-seawater interface when environmental conditions became favorable [16]. Despite the difficulties in fully understanding SIMCO biodiversity, it is commonly accepted that sea ice is characterized by highly active bacterial biomass [11]. Namely, the microbial activity in 1 m thick sea ice, compared to the integrated values for the underlying 100 m layer of water column [17], reaches its maximum at the bottom-most, warmest horizon of the sea ice due to its contact with seawater [18].

SIMCOs largely consist of a mixed population of autotrophic algae and heterotrophic prokaryotes, whose biomass production (BP) is based on sunlight and the assimilation of dissolved organic carbon (DOC), respectively. Although some studies have shown significant variations in BP values under conditions of high primary production, the bacterial carbon turnover typically increases during the sunlight season [18]. SIMCOs play a significant role in the polar carbon-cycle by enhancing primary sea ice production, with direct effects on the underlying water column. Microbes are directly involved in phosphate and nitrogen remineralization [19,20].

A comprehensive assessment of microbial diversity and community composition inhabiting Antarctic sea-ice was conducted through cultivation and cultivation-independent molecular techniques (see [18] for further references). Despite the difference in investigation approaches and/or sequencing technologies applied, most genera observed in SIMCOs have members common to surrounding marine environments; however, there are some phylotypes that appear to be unique or strongly favored within sea ice [18]. The overwhelming majority of bacteria isolated from polar sea ice belong primarily to three high taxa: the classes *Gammaproteobacteria* (the orders *Oceanospirillales* and *Alteromonadales*) and *Alphaproteobacteria* (the order *Rhodobacterales*), and the phylum *Bacteroidota* [15,21,22]. Notably, these cultivation studies demonstrated significant differences in SIMCO structures, recovered during winter and summer seasons. Namely, a strong dominance of true psychrophilic sea-ice-associated bacteria over psychrotolerant species was detected during the dark winter months, while the latter remain dominant during the biologically active sunlight months [22,23,24,25,26]. Such drifts in SIMCO structure were further confirmed by community fingerprinting and analyses of clone libraries for bacteria and archaea, obtained from young and mature sea ice [7]. Typical seawater phylotypes, including members of Marine Group I of *Thaumarchaeota* and the ubiquitous α-proteobacterial clade SAR11, were abundantly recovered in young sea ice samples, while they found to rarely be present in mature sea ice. These data suggest that seasonal change contributes greatly to shaping the sea-ice-associated microbial community via taxon-specific selection during ice formation in autumn, followed by taxon-specific mortality in the ice during the dark winter months.

In the framework of the Italian Research Programme in Antarctica (PNRA), a deep metabarcoding approach was applied to study SIMCOs thriving in the coastal annual pack ice of Terra Nova Bay (Ross Sea, Antarctica). In particular, two pack-ice horizons were examined together with the ice–water interface, consisting of platelet interstitial and under-ice seawater samples. An additional study was performed to investigate the bacterial succession in Antarctic sea-ice as it formed, and to evaluate the effect of the algae-derived DOM content in the parent ice–water interface matrices on the developing sea-ice bacterial communities. Special emphasis was paid to determining whether or not the young SIMCOs, formed during freezing, reflected under-ice seawater or annual pack-ice communities. Given that anaerobic bacterial communities, including potential sulphate-reducing and sulfur-oxidizing bacteria, were recently tracked in Antarctic winter pack ice [27], osmolyte-degrading enrichment experiments were also established and analyzed.

## 2. Materials and Methods

### 2.1. Site Description and Sampling

During the XXXI Italian Expedition to Antarctica (November 2015), we investigated the distribution and taxonomic composition of sea-ice-associated microbial communities residing the brine inclusions in the annual sea pack-ice. For these purposes, two cores (thickness 220 cm, 150 cm Ø) were pulled out close to each other from the coastal annual pack ice of Terra Nova Bay (Ross Sea, Antarctica: S 74°41.199′ E 164°07.117′), one nautical mile from the Italian base *Mario Zucchelli Station* (MZS). Using specially made sterile stainless-steel tubing and containers, brine was collected from the sibling cores by drilling incomplete holes and collecting the internal brine in a short time, similar to the process described elsewhere [28]. Two depths were chosen: the upper horizon (about 50 cm below the snow–ice interface, API_top_) and the lower horizon (about 50 cm above the ice–seawater interface, API_bottom_). After processing, the twin ice-cores were positioned horizontally on a wooden holder so that brine samples could be taken by gravity percolation in less than one hour. After this, approximately 1000 mL (4000 mL in total) of the brine was collected from each location of both ice cores. Using a conventional thermometer and refractometer Master-S28 OK (Atago, Japan), the brine temperature and salinity of both cores were measured directly at the sampling site. In addition, almost 2000 mL of platelet ice interstitial water (PLI) and 10,000 mL of under-ice seawater (UISW) were collected from the formed holes into two separate sterile plastic containers. All samples were immediately transported to MZS and processed within 6 h after sampling. Because both the temperature and salinity values were identical, brine samples collected from the same ice core horizons were pooled and filtered through 0.22 µm Sterivex filters (Durapore; Millipore, Billerica, MA, USA) using a peristaltic pumping system. The filters were then stored in lysis buffer (40 mM EDTA, 50 mM Tris/HCl, 0.75 M sucrose) at −20 °C. Before proceeding with the DNA extraction, performed in Italy, the Sterivex filters were first thawed on ice, then 400 μL of TE buffer (pH 8.0), containing lysozyme (5 mg mL^−1^) and proteinase K (0.2 mg mL^−1^, final concentrations), was added inside the cartridges; they were shaken for 5 s and incubated for 10 min at room temperature. A total of 1600 μL of QRL1 lysis buffer (containing β-mercaptoethanol) was added inside the Sterivex cartridges and extraction was performed using a Qiagen RNA/DNA Mini Kit (Qiagen, Milan, Italy), according to the manufacturer’s instructions. The DNA samples were further concentrated using a microconcentrator (Centricon 100; Amicon, Millipore, Billerica, MA, USA). The quantity, integrity and purity of DNA was checked and evaluated using both agarose gel electrophoresis, and a NanoDrop^®^ ND-1000 Spectrophotometer (Wilmington, DE, USA).

### 2.2. Simulation of Winter Freezing Process and Sea-Ice Brine Formation

In order to simulate the winter freezing process, which leads to the fast formation of young pack ice, 1500 mL of platelet-ice interstitial water and 9000 mL of under-ice seawater were transferred in a sterile transparent Plexiglas cylinder (150 cm high, 10 cm Ø, total volume of approximately 12,000 mL) and incubated for one week at −10 °C in the dark. After the formation of ‘simulated fast ice’ (SFI), the cylinder was transferred into a 2000 L aquarium and kept floating for one moth at a constant temperature of −2 °C in the dark. Using the holes previously made in the Plexiglas cylinder with intervals of 15 cm and plugged with rubber stoppers, the SFI was checked weekly for oxygen and salt concentration. After one month of incubation, the bottom-most part of the formed SFI was cut, with an ethanol-wiped handsaw, into six pieces (each horizon 15–20 cm, hereafter called B1–B6), crushed and melted at 2 °C in the dark for 12 h; after this, the rest of the ice was removed (Figure 1). The melted samples (approximately volume of 1 L each) were immediately filtered through 0.22 µm Sterivex filters (Durapore; Millipore, Billerica, MA, USA) using a peristaltic pumping system, and processed as described above.

### 2.3. Microbial Community Analysis

Libraries of the V3–V4 hypervariable regions of 16 S rRNA were prepared by single PCR using double-indexed fusion primers with heterogeneity spacers, described by Fadrosh et al. (2014) [29]. Part of the forward primers, annealing to 16 S rRNA, corresponded to the Pro341 F (CCTACGGGNBGCASCAG) primer [30]; reverse primers corresponded to modified the R806 prokaryotic primer (GGACTACHVGGGTWTCTAAT) [31]. Two nanograms of DNA were used for the reaction. Amplification was performed using qPCRmix-HS™ SYBR mastermix (Evrogen, Moscow, Russia) under the following conditions: 30 cycles of denaturation at 95 °C for 15 s; primer annealing at 58 °C, 15 s; and DNA synthesis at 72 °C, 25 s, followed by final incubation for 5 min at 72 °C. The qPCR was performed using a CFX96 real-time PCR instrument (Bio-Rad Laboratories, Hercules, CA, USA) and the analysis of the amplification curves of the samples, compared with negative controls, was used as an additional control point. Purification of PCR products was performed using the Cleanup Mini kit (Evrogen, Moscow, Russia), according to the manufacturer’s instructions. The quality of the final libraries was assessed using electrophoresis in 2% agarose gel, and quantification was performed with a Qubit™ fluorometer (Life Technologies, Carlsbad, CA, USA). After quantification, the libraries were pooled equimolarly.

Libraries were sequenced with MiSeq™ Personal Sequencing System (Illumina, San Diego, CA, USA) using paired-end 300-bp reads. Demultiplexing was performed using a publicly available pipeline, described previously [29], with minor modifications. Thus, just after the extraction of barcodes, primer sequences were trimmed from the reads using CLC Genomics Workbench v. 10.0 (Qiagen, Düsseldorf, Germany). After that, overlapping read pairs were merged using the SeqPrep tool (https://github.com/jstjohn/SeqPrep, accessed on 25 July 2017) with default parameters. The resulting read files were synchronized with index read files and demultiplexed.

### 2.4. Bioinformatic Analyses

Bacterial Tag-Encoded FLX Amplicon Pyrosequencing (bTEFAP) for Microbiome Studies [32] was applied to analyze the microbial composition. In all, approximately 230,000 raw reads, covering the V3–V4 hypervariable region of the 16 S ribosomal RNA gene, were obtained with the Illumina MiSeq platform. Sequences were depleted of barcodes and primers, then sequences <150 bp or with ambiguous base calls, and with homopolymer runs exceeding 6 bp, were removed. Demultiplexed sequences were further processed using the Quantitative Insight into Microbial Ecology (QIIME) v.1.9.1 open-source software package [33] using the following work flow [34,35]: Operational Taxonomic Units (OTUs) were picked at 97% similarity using “pick_open_reference_otus.py” (http://qiime.org/scripts/pick_open_reference_otus.html, accessed on 17 February 2022). This script clusters reads against the Greengenes v13_8 database [36]. Chimeric sequences were recognize by identify_chimeric_seqs.py (http://qiime.org/scripts/identify_chimeric_seqs.html, accessed on 17 February 2022) against the qiime-default-reference (85_otus.pynast.fasta). The OTUs were taxonomically classified using the Ribosomal Database Project Classifier [37] against a SILVA-based reference database (version v123) [38]. Samples were then rarefied using single_rarefaction.py (http://qiime.org/scripts/single_rarefaction.html, accessed on 17 February 2022) at a value of 10,298 sequences, based on the minimum number of sequences found in the most indigent sample in the dataset. For successive statistical analysis OTUs, sequences were used to construct a phylogeny tree using make_phylogeny.py (http://qiime.org/scripts/make_phylogeny.html, accessed on 17 February 2022) using FastTree [39] with default values.

### 2.5. Statistical Analyses

Within-community diversity (alpha diversity) was calculated using observed OTUs, Chao1 and Shannon indexes, with 10 sampling repetitions at each sampling depth, using alpha_diversity.py (http://qiime.org/scripts/alpha_diversity.html, accessed on 17 February 2022). The script beta_diversity_through_plots.py (http://qiime.org/scripts/beta_diversity_through_plots.html, accessed on 17 February 2022) was used to analyze statistical differences between samples (beta diversity) and generate Principal Coordinate Analysis (PCoA) plots. Hierarchical Cluster analysis (cluster mode, group average) was applied on a Bray–Curtis similarity matrix obtained from the rarefied OTU abundance table within PAST PAleontological STatistics V3.25 (https://palaeo-electronica.org/2001_1/past/issue1_01.htm, accessed on 17 February 2022). The same OTU table and the same program were used to calculate the contribution of single OTUs to the observed difference between samples (SIMPER test) and the ANOSIM (ANalysis Of SIMilarities) [40] to assess the abiotic factor that mainly contributed to shaping the microbial communities.

### 2.6. Sequencing Data

All sequence data obtained in this study are freely available at the European Nucleotide Archive (ENA)/NCBI under the accession number PRJNA807589.

### 2.7. Enrichment Experiments

Brine samples obtained from the API_top_ and API_bottom_ layers of Terra Nova Bay annual pack ice were used for the enrichment and cultivation of anaerobic microorganisms that decompose compatible solutes. For this purpose, the ONR7 a mineral basic medium [41] was adjusted to a salinity corresponding to each brine sample (43 psu and 78 psu) by addition 5.0 g L^−1^ and 40 g L^−1^ NaCl, respectively. After autoclave sterilization, 1 mL L^−1^ of acidic trace metal solution, 1 mL L^−1^ of vitamin mix [42], alkaline Se/W solution [43] and 50 mg L^−1^ of yeast extract were added to the ONR7, a base media. For enrichment with psychrophilic anaerobes that decompose compatible solutes, stock solutions of choline and dimethyl sulfide (DMS) (2 M in each case, both from Sigma Aldrich Merck KGaA, Darmstadt, Germany) were filter sterilized and added to medium at a final concentration of 20 mM. Serum bottles of 120 mL were filled with 90 mL of sterile medium, and after adding 10 mL of each brine sample, anaerobic conditions were achieved by the addition of 0.05 mM cysteine (Sigma Aldrich, Merck KGaA, Darmstadt, Germany). The bottles were further subjected to three cycles of evacuation/flushing with sterile argon. Incubation was carried out at 0 °C in the dark for 6 months without shaking. All cultivations were performed in triplicates.

## 3. Results and Discussion

### 3.1. Basic Physico–Chemical Analysis

With direct field measurements of temperature and salinity, the brine samples taken from the same horizons of twin cores turned out to be identical (−7.1 °C, 43 psu for the API_top_ horizons, and −4.7 °C, 78 psu for the API_bottom_ horizons). Considering that, at the time of sampling, the station was covered with a thick layer of snow (mean 12.5 cm), efficiently isolating the pack ice from cold air, but not from solar heating, meant that such differences in temperature measured at different depths of pack ice were evident. At both depths, the ice was permeable with a relative brine content of approximately 3–5% calculated as a function of temperature and salinity [28]. As mentioned elsewhere [27], the permeability of the pack ice provides brine transport, and therefore, potentially favorable conditions for bacteria and sea ice algae. Indeed, both horizons were found to be well oxygenated (Table 1), which can be explained by the presence in the pack ice of photosynthetic diatoms of the genera *Amphiphora* and *Nitschia*, further visualized in the laboratory using microscopy analysis of the brine samples. On the contrary, with salinity and pH ranges close to those of pack ice, all the analyzed horizons of the simulated fast ice represent a micro-oxic environment with an oxygen content of no more than 3.1 mg L^−1^. This depletion of oxygen was most likely associated with the exposure of fast ice to the dark, which stimulated algae death, followed by their decomposition by aerobic bacterial heterotrophs, which is an oxygen-consuming process. The bottom-most SFI layers were more oxygenated than the upper horizons, probably due to their higher salinity, which somehow interfered with the metabolic activity of the bacteria.

### 3.2. Alpha and Beta Diversity, Richness Metrics

As noted in the Materials and Methods section, microbial diversity was studied in different matrices: (i) brine samples taken from different horizons of annual pack ice (API_top_ and API_bottom_, respectively); (ii) platelet ice interstitial water (PLI); (iii) under-ice seawater, collected directly below pack ice (UISW); and (iv) six layers of simulated fast ice (SFI_L1_-SFI_L6_), taken from the bottom to the middle part of the core (Figure 1). After quality control and data filtering, a total of 228,812 reads were obtained. Approximately 10% of these reads were removed by CHECK_CHIMERA analysis [44], and all resulting rarefaction curves showed gentle slopes, indicating that deep sequencing was sufficient to capture the diversity of microbial communities from both natural matrices and fast simulated core samples (Appendix A).

To study the prokaryotic diversity and richness in these samples, unique amplicon sequence variants (ASVs or OTUs), ACE, Chao1, Shannon, Simpson and Dominance indexes were calculated. The number of species observed indicates that the greatest diversity was found in natural samples, regardless of whether they were less (UISW) or more saline (API_bottom_) (Table 1 and Table 2).

The smallest number of ASVs and, accordingly, the least prokaryotic diversity were found in all SFI samples, with a general trend of decreasing diversity towards the middle horizons of the analyzed SFI core (Table 1). As already noted for natural samples, the least and most saline analyzed SFI layers had the highest number of ASVs, and thus, the highest prokaryotic diversity. Notably, this trend replicated the oxygen depletion profile, with an oxygen minimum of 1.9 mg L^−1^ found in the SFI_L3_ horizon.

As above, beta diversity analyses showed a clear separation of microbial biodiversity between natural samples and simulated fast-ice samples. The unweighted uniFrac distance takes into account the presence and absence of different ASVs along with their phylogenetic distance, while the weighted uniFrac distance also considers the relative abundance of the different ASVs identified in each sample [45]. The difference in methodologies may explain the difference between the outcomes of the principal coordinate analysis (PCoA) derived from the weighted and unweighted uniFrac distance matrix. However, a clear difference between the microbial communities found in natural samples and those found in SFI samples can be seen in both PCoA plots (PC1 36.55 and 57.38%, respectively). Notably, oxygen concentration appears to be a more important factor than salinity in microbiome shaping across different samples. Three different groups were formed depending on the oxygen content: (i) well-oxygenated natural samples; (ii) the most saline bottom horizons SFI_L1_ and SFI_L2_ with an oxygen content of 2.3–3.1 mg L^−1^; and (iii) the most oxygen-depleted upper horizons of the SFI core, respectively (Figure 1 and Figure 2). A similar grouping of microbial communities was obtained using cluster analysis, which considers only the relative contribution of each ASV without taking into account phylogenetic affiliation (Figure 2). In addition, the contribution of both salinity and oxygen to the formation of microbial biodiversity was analyzed using the ANOSIM test, and only oxygen showed a statistically acceptable effect (R = 0.4231, *p* = 0.0165).

### 3.3. Microbial Diversity in Natural Samples

A SIMPER test [40] was applied, to examine the contribution of the most abundant microbial species, to the Bray Curtis matrix used for cluster analysis. According to the SIMPER test, under-ice seawater, platelet ice and both pack-ice brine samples were characterized by an abundant fraction of chloroplast-derived small-subunit (SSU) rDNA gene sequences (14.45 Av. dissim.). These sequences covered from 65.74% of all PLI reads to 14.13% of the sequences found in API_top_ pack ice brine. UISW and API_bottom_ were intermediate, with 48.51% and 44.87% chloroplast-derived SSUs among all the reads analyzed, respectively (Figure 3 and Figure 4; Appendix A). This high abundance of algal reads is typical of microbial communities associated with sea ice during a solar period [15]. They were mainly derived from the diatoms *Amphiprora* sp. and *Nitschia* cfr. *stellata*, which are known to dominate sympagic communities at the bottom of annual pack ice in Terra Nova Bay [10]. The presence of these phototrophic organisms, even in the upper layer of the pack ice, was not unexpected, since many pennate forms of sea ice algae are able to move through the brine network by releasing sticky extracellular polymeric substances [46].

In addition to diatoms, the composition of prokaryotes found in all samples, collected by us in Terra Nova Bay, was also very similar at the high taxonomic level. The phylum Proteobacteria (represented by the classes Alpha- and Gamma-proteobacteria) and the phylum *Bacteriodetes* were the dominant taxa, as previously reported for microbial communities associated with Antarctic first-year ice and multi-year ice [15,21,22,27,47,48,49,50,51,52,53,54]. However, the prevalence of the most abundant genera varied among the UISW, PLI and the pack ice brines, showing a shift from predominance of common seawater bacteria, such as *Colwellia* and *Sulfitobacter*, to more typical sea-ice bacteria such as *Paraglaciecola*, *Loktanella* and *Polaribacter* (Figure 3 and Figure 4).

After SIMPER analysis at the genus level, the two dominant *Gammaproteobacteria* ASVs, namely the genera *Pseudomonas* (2.838 Av.dissim.) and *Paraglaciecola* (1.054 Av.dissim.), covered 22.23% and 9.61% of all API_bottom_ and API_top_ SSU gene reads, respectively. The atypical prevalence of *Pseudomonas*-related reads can be somehow explained by the presence of a significant amount of *Pleuragramma antarctica* (silverfish) eggs observed at the bottom of both pack ice cores during brine sampling. Indeed, the most similar (98.54% of identity) SSU gene sequence, found in the NCBI database, belongs to *Pseudomonas* sp. S8 (accession number KT223376), isolated from the eggs of Atlantic salmon (*Salmo salar*) [55]. Representatives of the genus *Colwellia* dominated the UISW bacterial community, accounting for 9.57% of all reads, while their contributions to microbial diversity in other natural samples were minimal (less than 0.5%). Notably, despite covering nearly 6% of all reads, no dominant group (≥2%) among the 59 ASVs affiliated with the class *Gammaproteobacteria* was found in the platelet-ice interstitial water (Appendix A). As previously mentioned, a *Bacteroidetes* ASV classified as a member of the genus *Polaribacter* (2.76 Av.dissim. with 26.26% of all SSU gene reads), and an alpha-proteobacterial ASV belonging to *Loktanella* (1.923 Av.dissim., 17.16% of all reads)—both typical Antarctic bacteria associated with diatom bloom [27]—were among the most dominant microbial groups in the upper pack-ice brine, outperforming the dominance of chloroplast (14.13%) observed in other natural matrices.

### 3.4. Microbial Diversity in Simulated Winter Ice

In contrast to spring and summer, the activities of sea-ice bacteria during the Antarctic winter are poorly understood, and their community dynamics have not been yet described, since the information available is limited to isolated bacterial strains [22]. In order to somehow overcome the serious logistical difficulties associated with organizing the sampling of pack ice during the dark and very cold Antarctic winter—and especially given the lack of a permanent year-round scientific station in Terra Nova Bay—we conducted a laboratory experiment simulating the rapid formation of young winter ice. As described in the relevant Materials and Methods and Results sections, by applying our experimental approach using samples of platelet ice and free-ice seawater after one month of exposure, we were able to recreate a 150 cm-high fast ice core showing the three-dimensional complexity of ice-pore brine of varying salinities and oxygen contents (Figure 1). The stratification of the brine with increasing salinity towards the bottom was a good approximation to natural conditions, since similar salinities of ice-pore brines were observed in the analyzed samples of annual pack ice in Terra Nova Bay (Table 1). However, unlike these natural brines, as we expected, all analyzed samples turned out to be micro-oxic, with an oxygen content less than a quarter of that measured at the beginning of the experiment.

As mentioned above, the simulated fast ice showed low microbial diversity, with the number of species being half that observed in natural samples (Table 2). Besides this, the results of the phylogenetic analysis also showed that the simulated wintertime microbial community remarkably differed from the algae-dominated autumnal sea-ice communities (Figure 4). One of the most important differences was the absence of PCR-amplifiable DNA (SSU gene amplicons) belonging to diatoms, which indicated the complete degradation of algal DNA by SFI microbial communities that occurred in just one month of the experiment. Such a rapid conversion of algal DNA was not entirely unexpected, since evidence of the ability of marine bacteria to rapidly mineralize algal exudates appeared as early as 1933 [56]. Notably, more recent data showed that microbial activity against algae-derived organic matter was limited to dead diatoms and did not include actively growing cells [57,58,59]. It was also demonstrated that during the initial period of quick bacterial decomposition of algae, their intracellular components from the cytoplasm, including nucleic acids, were primarily consumed, which led to the accumulation of membrane-associated proteins; a third of these, as shown, can also be cleaved within a very short time, without preferential degradation of the overall protein pool [60]. It can be reasonably assumed that it was this very rapid decomposition by copiotrophic aerobic bacteria of algae-derived organics that led to the observed formation of an oxygen-depleted environment and to the corresponding shift towards a community consisting mainly of microaerophilic, facultative, and even strictly anaerobic organisms (Figure 4).

Among the dominant bacteria (accounting for at least 2% of all SFI reads), only Gammaproteobacteria of the genus *Colwellia* were also detected in the initial seawater, recruiting almost a tenth of the UISW reads. However, few of these sequences were found also in platelet ice and annual pack-ice samples. Interestingly, *Colwellia* dominated the saltiest SFI_L1_ and SFI_L2_ bottom horizons (26.81 and 46.06% of all reads, respectively) and were found to be 99% identical to organisms previously found in bottom ice during the sea ice formation experiment, conducted in a mesocosm with organic-enriched waters of the North Sea [61]. In addition to broad salt tolerance and the ability to grow at subzero temperatures, representatives of the *Colwelliaceae* family are characterized as facultative anaerobic bacteria [62], which can explain their presence in both well-oxygenated natural and micro-oxic SFI samples.

Of special interest was the finding of *Oleispira antarcica*-related ASVs in all analyzed SFI samples. Whereas only a few of such sequences (<0.04%) were found in the original seawater and platelet ice, they peaked at 23.95% of all reads derived from the upper SFI_L6_ layer. These organisms belong to the group of obligate marine hydrocarbon-degrading bacteria [63,64,65] and their predominance in a simulated winter-ice core can likely be explained by the presence of aliphatic C_15_-C_18_ hydrocarbons released by decaying diatoms [66,67].

Apart from these bacterial taxa, the SFI ice samples showed a predominance of atypical sea-ice bacterial groups, represented by facultatively anaerobic *Epsilonproteobacteraeota* (genera *Arcobacter*, *Sulfurimonas* and *Sulfurospirillum*); *Bacteroidia* (genus *Marinifilum*); and, quite unexpectedly, by strict anaerobic microorganisms belonging to *Deltaproteobacteria* (genus *Desulforhopalus*), *Fusobacteria* (genus *Psychrilyobacter*) and *Clostridia* (genera *Caminicella*, *Clostridiisalibacter* and *Fusibacter*). Since they all have different metabolic preferences, their uneven distribution over different horizons of the simulated winter-ice core may indirectly indicate the predominance of one type of catabolism or another occurring there. Namely, the less saline (≤51 psu) SFI layers were dominated by both sulfate-reducing *Desulforhopalus* and sulfur-oxidizing ε-proteobacteria, suggesting that, under suitable conditions, sulfur-containing compounds may be actively transformed in winter sea-ice. In addition to seawater, sulfate and sulfur-containing amino acids; dimethylsulfoniopropionate (DMSP), a known compatible solute, produced in large quantities by polar ice diatoms as an osmo- and cryoprotectant [68,69,70]; and climatically active dimethylsulfide (DMS), resulting from the enzymatic cleavage of DMSP, are the most likely sources of such compounds. This reaction can be carried out by ε-proteobacteria, since *dddY*-like genes, encoding periplasmic DMSP lyase, were found in various *Arcobacter* strains [71]. Moreover, it has been demonstrated that DMSP and DMS, present in low quantities in marine sediments, can be degraded by sulfate reducers, resulting in the formation of CO_2_ and H_2_ S as the end-products of their oxidation [72]. Thus, based on the above, biogenic sulfur cycling appears to be very active in the middle zones of the simulated winter-ice core, as the density of sulfur-oxidizing ε-proteobacteria can reach 60.57% of all reads, obtained from the less saline SFI_L6_ layer analyzed (39 psu). Representatives of the genus *Desulforhopalus* were found in the next three lower layers and reached their maximum (23.97% of all reads) in SFI_L5_, indicating an active sulfate reduction there.

In turn, the most saline near-bottom part of the SFI core was a zone predominantly enriched in hydrolytic, and fermenting strictly anaerobic psychrotolerant representatives of the genus *Psychrilyobacter* (38.32% and 29.89% of all SFI_L2_ and SFI_L3_ reads, respectively) and spore-forming clostridia of the genus *Caminicella* (43.61% of all SFI_L3_ reads). The discovery of representatives of the latter genus among the main microorganisms involved in the rapid mineralization of organic matter was intriguing. Although originally described as thermophilic, they are often found on the cold seafloor of the Arctic [73]. In addition, we are aware that cultivation-free approaches such as high-throughput SSU gene-sequencing are likely to be more challenging methods of obtaining insight into the metabolism of microorganisms associated with Antarctic sea-ice, as this specific environment may contain undocumented microbial strains that lack reference genomes. Moreover, it is well known that closely related bacterial strains with nearly identical SSU genes can have very different genomic DNA content, as has been demonstrated for common sea-ice bacteria of the genus *Octadecabacter* [74], meaning that we may miss a full understanding of their encoded metabolic preferences [27].

To our knowledge, apart from previous evidence of transient anoxic conditions and anaerobic reactions such as denitrification in the Arctic ice [75], there is only one report describing an active anaerobic microbial community thriving in Antarctic winter pack-ice [27]. A predominance of sulfate-reducing *Desulforhopalus*, along with sulfur-oxidizing *Arcobacter* and *Sulfurospirillum*, was observed in the middle section of a winter pack-ice core taken from the Weddel Sea, which had a strong hydrogen sulfide odor indicative of anoxic conditions. Unfortunately, based on the limited current data, we are not only unable to deduce how common anaerobic bacteria are in Antarctic winter pack-ice, but we are also unable to draw a conclusion about their origin. However, firstly, the presented data indicate that, under certain conditions, these key anaerobic bacteria can become locally predominant in winter pack-ice; secondly, such a striking similarity of the structures of the microbial community in the anoxic pack ice of Weddel Sea, with the community observed by us in the SFI, as the proof of concept, confirms the feasibility of our method for modelling winter sea-ice formation.

### 3.5. Anaerobic Enrichments

Many bacteria and diatoms associated with sea ice are adapted to thrive in sea-ice brines by producing increased amounts of secondary metabolites known as osmolytes [76]. When stored at a high enough concentration, they also provide some degree of freeze protection [77]. These osmolytes are short-molecular-weight methylated compounds containing either sulfur or nitrogen in their backbone. Among them, glycine betaine (*N*,*N*,*N*-trimethylglycine) and DMSP, mentioned above, are the most common and widespread in the marine environment [78]. Considering that these osmolytes can be a source of carbon, energy and nutrients for the microbial community of Antarctic winter pack ice, we used a combination of choline (a derivate of glycine betaine) and DMS (a derivate of DMSP) to enrich the API_top_ and API_bottom_ brine samples. Given that anaerobic bacterial communities, including potential sulphate-reducing bacteria, were recently tracked in Antarctic winter pack-ice [27], special attention was paid to the establishment of similar communities in our enrichment experiments. After 6 months of incubation, strong turbidity was observed in both types of enrichment cultures. However, only the less saline API_top_ enrichment (43 psu) turned black and gave off a strong smell of hydrogen sulfide, indicating the growth of sulfate reducers. For this reason, only this one was further processed for high throughput sequencing of SSU genes.

The results of this analysis indicated that the microbial community of API_top_ enrichment was dominated by sulfate-reducing bacteria belonging to the genera *Desulforhopalus* (36.82% of all reads) and *Desulfosporosinus* (5.33%), followed by the facultatively anaerobic fermenting *Marinifilum* (18.23%), sulfur-oxidizing ε-proteobacteria of the genus *Arcobacter* (11.42% of all reads) and strictly anaerobic psychrotolerant representatives of the genus *Psychrilyobacter* (6.99% of all reads). Although the spore-forming clostridia of the genus *Caminicella* do not belong to the group of dominating organisms (recruitment ≥2% of all reads), they were also present in enrichment, accounting for 1.54% of all sequences. Is it worth noting that all of these dominant microorganisms appear to represent the so-called ‘dark part’ of the pack-ice microbial community, as they were not detected by Illumina’s deep sequencing of the original brine samples. It can be seen from this list that our enrichment approach created a microbial community strikingly similar to those we observed in the upper layers of the simulated winter ice, despite the fact that they originated from different natural sources (brine of annual pack ice and under-ice matrices, respectively). Following this finding, we can state that the Antarctic pack-ice and surrounding ambient are likely to be very dynamic and can quickly respond to environmental changes caused by the seasonal fluctuations, such as lack of sunlight in winter. Given the size of the Antarctic pack-ice, even in total darkness and cessation of photosynthesis, its ecosystem appears to remain active, continuing to take part in global carbon and sulfur cycling.

## 4. Conclusions

Our late spring study of the microbial community inhabiting the annual pack ice of Terra Nova Bay is consistent with most studies conducted during the same period. It has been shown that widespread sea-ice-associated copiotrophic bacteria of the genera *Polaribacter* (the phylum *Bacteriodetes*) dominate the bacterial communities along with copiotrophic Gammaproteobacteria (the genera *Paraglaciecola* and *Pseudomonas*) and Alphaproteobacteria (the family *Rhodobacteraceae*). It is well known that sea-ice microbial community is highly dependent on a number of abiotic factors such as temperature, salinity and, in particular, light and oxygen content [5,7,27,79,80,81,82,83,84]. Because these factors vary greatly with the seasons, they certainly also cause seasonal dynamics in bacterial abundance and community composition in sea ice. However, compared with numerous studies carried out during solar seasons, there are very few studies related to winter observations of the microbial community dynamics. To fill this gap, we conducted ice formation experiments in the laboratory along with a cultivation approach. The simulation experiment was designed primarily to reproduce the two critical conditions faced by the pack-ice-associated microbial community during winter—namely complete darkness, causing the cessation of photosynthesis, and the presence of a large amount of algae-derived organic matter accumulated during the light seasons in the ice. It was assumed that in the absence of photosynthesis, the activity of aerobic copiotrophs could most likely lead to the creation of a micro-oxic or even strictly anoxic environment, which would cause the emergence of new players, namely facultative and anaerobic microorganisms. To understand such dynamics, the structure of the bacterial community observed in different layers of simulated winter ice was compared with the structure of the community inhabiting natural annual pack ice, which made it possible to at least approximately estimate the biogeochemical role played by sea-ice-associated bacteria during the light and dark seasons. Using our approach, we managed to obtain, in a microbial community, the dominance of groups atypical for summer pack ice, namely: facultative anaerobic *Epsilonproteobacteraeota* (genera *Arcobacter*, *Sulfurimonas* and *Sulfurospirillum*); *Bacteroidia* (genus *Marinifilum*); and strictly anaerobic microorganisms belonging to *Deltaproteobacteria* (genus *Desulforhopalus*), *Fusobacteria* (genus *Psychrilyobacter*) and *Clostridia* (genera *Caminicella*, *Clostridiisalibacter* and *Fusibacter*). The structure of the SFI microbial community was strikingly similar to that obtained by us in anaerobic enrichment with the addition of compatible solutes (choline and DMS). This finding suggests that both oxygen depletion and the presence of algae-derived organic compounds are likely key factors that shape the microbial population in sea ice during the dark winter season.

As a general outcome, it seems that in spite of harsh environmental conditions appearing during the dark, cold winter months, the microbial community of Antarctic pack-ice continues to operate and participate in global carbon and sulfur cycling. Overall, our results indicate that sea-ice-associated microbial communities can remain dynamic throughout winter if physical conditions are favorable, and if the algal-derived organics are high enough to support the activity of both aerobic and, as we found out, anaerobic bacteria.

## Figures and Tables

**Figure 1 microorganisms-10-00623-f001:**
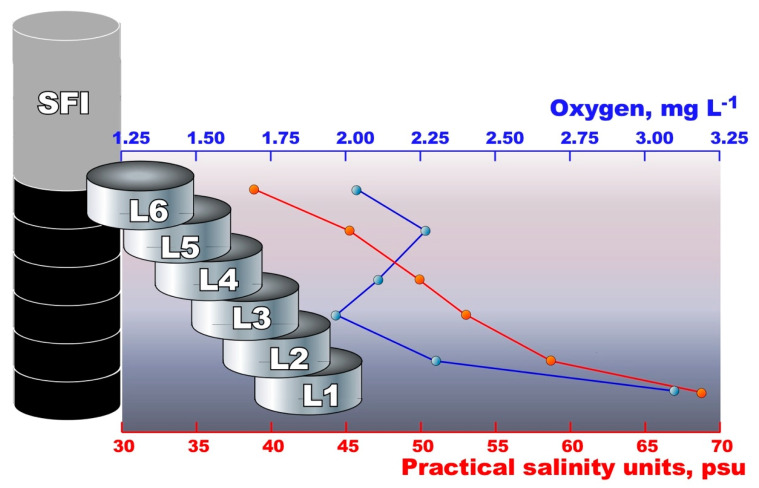
Salinity and oxygen concentration profiles observed in the lower six sections of the simulated fast ice core (SFI) after a month in the dark at −2 °C. The data are mean values from three parallel measurements.

**Figure 2 microorganisms-10-00623-f002:**
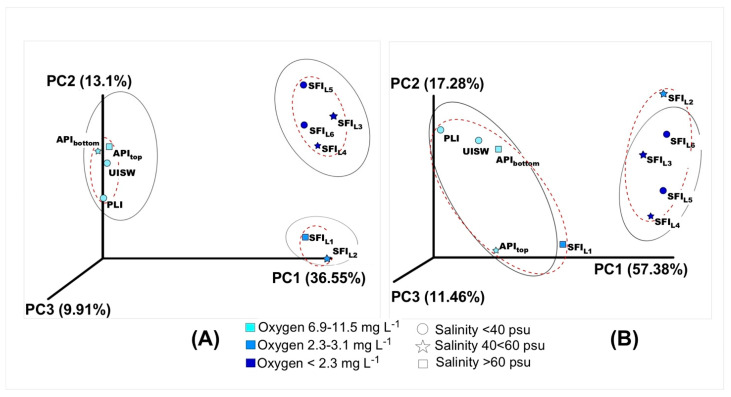
Principal component analysis (PCoA) of Antarctic sea-ice-associated microbial communities: (**A**) unweighted UniFrac, (**B**) weighted UniFrac. Abbreviations used: API—annual pack ice; PLI—platelet-ice interstitial water; UISW—under-ice seawater; SFI_L1−6_—different layers of simulated fast ice.

**Figure 3 microorganisms-10-00623-f003:**
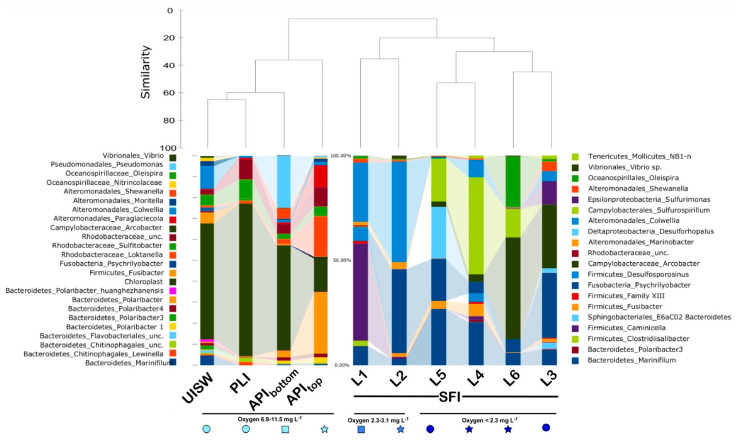
Histograms of bacterial genera and chloroplast-related SSU rDNA genes identified in microbial communities associated with Antarctic sea-ice. ASVs which could not be resolved at the genus level were reported with the designation ‘unc’ after the name of the nearest known parental rank. The list of the identified genera and their relative abundances is reported in Appendix A. The abbreviations used are similar to those shown in Figure 2.

**Figure 4 microorganisms-10-00623-f004:**
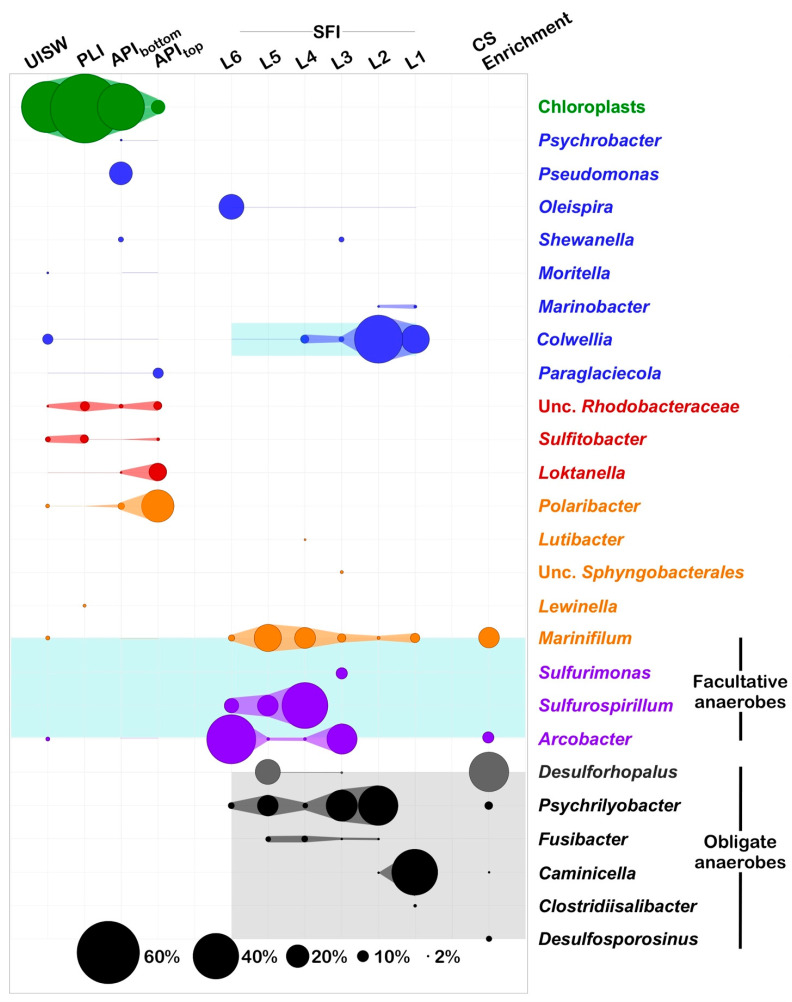
Relative abundances of bacterial genera and chloroplast-related SSU rDNA genes identified in microbial communities associated with Antarctic sea-ice, in simulated fast ice, and in anaerobic enrichment decomposing compatible solutes (CS Enrichment). The taxonomy is based on SILVAngs, release 123.1. Sequence abundances are given in percentage of total number of classified reads. Only taxa that made up >2% of the total sequences in any given sample are shown. The abbreviations used are similar to those shown in Figure 2 and Figure 3. A list of the identified genera and their relative abundances in CS Enrichment is reported in Appendix A.

**Table 1 microorganisms-10-00623-t001:** Main physico–chemical parameters in natural matrices and simulated fast ice brine samples.

Sample	Salinity, psu	Oxygen, mg L^−1^	pH	Temperature, °C	Redox, mV
Annual Pack Ice, Platelet Ice and Seawater					
API_top_	43	6.93	7.5	−7.1	ND
API_bottom_	78	7.36	7.8	−4.3	ND
PLI	35	11.52	8.3	−1.8	ND
UISW	34	8.80	8.1	−1.8	ND
Simulated Fast Ice					
SFI_L6_	39	2.04	8.0	−2.0	+108.0
SFI_L5_	45	2.26	8.0	−2.0	+101.4
SFI_L4_	49	2.10	7.6	−2.0	+103.2
SFI_L3_	53	1.96	7.7	−2.0	+91.3
SFI_L2_	58	2.33	7.5	−2.0	+91.8
SFI_L1_	68	3.10	7.3	−2.0	+86.3

**Table 2 microorganisms-10-00623-t002:** Prokaryotic richness and diversity estimates, based on 97% ASV clusters of Antarctic sea-ice-associated microbial communities. The diversity indexes: abundance-based coverage estimator of species richness (ACE), abundance-based estimator of species richness (Chao1), estimators of species richness (Shannon) and species evenness (Simpson).

Sample	Chimera Check		ASV	ACE	Chao1	Shannon	Simpson	Dominance
	Before	After						
Annual Pack Ice, Platelet Ice and Seawater								
API_top_	25,950	24,602	692	1376.11	1302.66	5.411	0.914	0.086
API_bottom_	24,889	22,407	768	1470.22	1305.66	4.793	0.814	0.186
PLI	27,614	26,913	541	959.96	925.17	5.252	0.930	0.070
UISW	25,127	23,583	1057	1943.7	1745.59	6.593	0.956	0.044
Simulated Fast Ice								
SFI_L6_	24,132	23,307	461	726.08	687.78	4.716	0.885	0.115
SFI_L5_	29,267	28,377	227	403.08	389.03	3.353	0.832	0.168
SFI_L4_	16,334	14,335	350	550.38	554.19	3.833	0.788	0.212
SFI_L3_	10,472	10,298	174	258.43	260.89	3.034	0.762	0.238
SFI_L2_	19,822	11,586	310	421.15	388.76	3.384	0.743	0.257
SFI_L1_	25,205	18,856	445	768.55	749.55	4.236	0.808	0.192

## Data Availability

Sequence data are available at the European Nucleotide Archive (ENA)/NCBI, study accession number PRJNA807589.

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
