# Peer review of "Wintertime Simulations Induce Changes in the Structure, Diversity and Function of Antarctic Sea Ice-Associated Microbial Communities"

_microorganisms, 2022, doi:10.3390/microorganisms10030623_

Round 1
Reviewer 1 Report
This study examines the succession of microorganisms in artificially constructed sea ice in the polar nights and how they affect the ecosystem of the Southern Ocean.
The most distinctive feature of the Southern Ocean is the repeated annual formation and melting of sea ice. As the authors have stated, it is very difficult to investigate the ecosystem in the Southern Ocean during the winter, which is closed to ice, due to the difficulty of setting up observation equipments. However, investigating what is happening during this period is important for a comprehensive understanding of the biological and physical phenomena in this area. Observations using mooring and drifting systems have gradually revealed biological and physical phenomena in seawater, but the dynamics of biological communities in sea ice are not yet well understood.
This study is an attempt to estimate this point by a kind of simulation (and also by enrichment cultivation experiments), and in fact it seems to have succeeded to some extent. The experimental results and discussions presented are very interesting. In the introduction, the previous research is fully introduced. The method and its description are also appropriate. The results and discussions are well and adequately described. Considering the minor points below, including just my question, I think it's suitable to be published in this journal.
1. L.70: Add “(BP)” after “biomass production”.
2. L.150-154: The description of DNA extraction is a bit vague as follows;
1) L.150-151: Were the Sterivex filters removed from the cartridge and immersed in the lysis buffer? Or did you put the lysis buffer in the cartridge?
2) L.152-153: Did lysozyme and protease treatment also take place in the cartridge?
3. L.182-185: I think this experiment is endpoint PCR, but why did you use the q-PCR kit?
4. L.185: The MiSeq sequencing requires a PCR to add the index sequences after the first PCR, and the conditions should also be noted.
5. L215-220: Isn't the font style wrong?
6. Table 2: Does “Observed species” mean the OTU? If so, write OTU, or mention somewhere that you considered the number of OTUs as observed species.
7. Figure 2: The letters in the legend should be a little bigger.
8. L421-427: This result and discussion is also very interesting to me. However, is the diatom (DNA) that was detected a lot in natural samples decomposed to the extent that it is not really detected? Is there any reference that could be helpful in that bacteria rapidly degrade diatoms, even if the conditions are not similar?
9. L443-444: Please show with the literature what kind of hydrocarbons are produced by the decomposition of diatoms by bacteria.
10. L496-499: Isn't the font style wrong?
Author Response
First of all, we would like to express our gratitude for positive evaluation of our manuscript.
Responses to the Reviewer#1.
- L.70: Add “(BP)” after “biomass production”.
ïƒ Added
- L.150-154: The description of DNA extraction is a bit vague as follows: i) L.150-151: Were the Sterivex filters removed from the cartridge and immersed in the lysis buffer? Or did you put the lysis buffer in the cartridge? ii) L.152-153: Did lysozyme and protease treatment also take place in the cartridge?
ïƒ Thanks for pointing on this issue. We changed this part of the Materials and Methods and added more detailed description of DNA extraction protocol (lines 171-181 in the modified version of manuscript).
- 3. L.182-185: I think this experiment is endpoint PCR, but why did you use the q-PCR kit?
ïƒ qPCR allowed us to analyze the effectiveness of amplification compared with negative controls (by Cq analysis) and served as an additional control point. To clarify this, we added this sentence (lines 227-229 in the modified version of manuscript):
“The qPCR was performed using CFX96 real-time PCR instrument (Bio-Rad, USA) and the analysis of amplification curves of samples, compared with negative controls, was used as an additional control point.
- L.185: The MiSeq sequencing requires a PCR to add the index sequences after the first PCR, and the conditions should also be noted.
ïƒ In our case we used a one-step PCR protocol and primers included both 16SrRNA annealing part and index sequences. The protocol we used is in the full accordance with the protocol, published by Fadrosh and colleagues [29] (https://microbiomejournal.biomedcentral.com/articles/10.1186/2049-2618-2-6).
- L215-220: Isn't the font style wrong?
ïƒ Corrected
- Table 2: Does “Observed species” mean the OTU? If so, write OTU, or mention somewhere that you considered the number of OTUs as observed species.
ïƒ Yes, observed species mean the OTU or unique amplicon sequence variants (ASV) with cut off value of 97% of identity. We corrected this in modified Table 2
- Figure 2: The letters in the legend should be a little bigger.
ïƒ Done
- L421-427: This result and discussion is also very interesting to me. However, is the diatom (DNA) that was detected a lot in natural samples decomposed to the extent that it is not really detected? Is there any reference that could be helpful in that bacteria rapidly degrade diatoms, even if the conditions are not similar?
ïƒ Thanks for this comment. Indeed, we were also surprised by those who quickly algal biomass was degraded, and that at the end of the experiment we found no traces of PCR-amplifiable algal DNA. We have expanded this part of the conclusion and added 5 more references describing rapid degradation of algal biomass by marine bacteria (see lines 510-526 in modified manuscript).
- L443-444: Please show with the literature what kind of hydrocarbons are produced by the decomposition of diatoms by bacteria.
ïƒ We added both type of hydrocarbons and 2 corresponding references, as requested (see lines 547-548 in modified manuscript).
- L496-499: Isn't the font style wrong?
ïƒ Corrected
Reviewer 2 Report
The presented manuscript "Winter time simulations induce changes in the structure, diversity and function of Antarctic sea ice-associated microbial communities" is a significant contribution to the understanding of microbial processes occurring in ice ecosystems. The current state of the issue is well described and provided with proper references. The experimental part was carried out at a high methodological level. The results obtained by the authors are well presented in graphs and diagrams.
Some recommendations and remarks are as follows:
- Since the text describes studies of microbial communities in different layers of ice, it would be reasonable to present a scheme/picture of their location as well as indicate the sampling sites. This could greatly facilitate the understanding of the text described.
- Additionally, it could be also reasonable to give a scheme of the modelling experiment and indicate the taken samples (in abbreviations) used further for analysis and mentioned in Tabl.1.
- Can the authors conclude that methylotrophic methanogens are totally absent in the samples with compatible solutes? Have experiments been carried out to enrich C1-methanogens using dimethylsulfide as one of their characteristic substrates?
- The text in Conclusions is a bit vague. In the first part of the section, the authors indicate the correspondence of the results with the already available data and explain the need for the study. In the second part, the authors describe the results in general terms. The article could benefit if all the results described in the text would be more clearly formulated, the most revealing data could be indicated and the main contribution of the work to the problem under study would be well defined.
Author Response
First of all, we would like to express our gratitude for positive evaluation of our manuscript.
Responses to the Reviewer#2.
Some recommendations and remarks are as follows:
- Since the text describes studies of microbial communities in different layers of ice, it would be reasonable to present a scheme/picture of their location as well as indicate the sampling sites. This could greatly facilitate the understanding of the text described.
ïƒ Thanks for this comment. There appears to have been an error in the uploading procedure as we prepared and submitted a “Graphical Abstract” accurately describing the location of the sampling site, the annual pack ice sampling operation, and a scheme/illustration of the experimental design. We will upload this drawing again.
- Additionally, it could be also reasonable to give a scheme of the modelling experiment and indicate the taken samples (in abbreviations) used further for analysis and mentioned in Tabl.1.
ïƒ See comment above. We explained all abbreviation used in the “Graphical Abstract”, which we will upload again.
- Can the authors conclude that methylotrophic methanogens are totally absent in the samples with compatible solutes? Have experiments been carried out to enrich C1-methanogens using dimethylsulfide as one of their characteristic substrates?
ïƒ Thanks for this comment. Indeed, we expected that enrichment supplemented with DMS could somehow stimulate the growth of methylotrophic methanogens. But NGS sequencing using universal primers (amplifying both bacteria and archaea) revealed a negligible presence (less than 0.01% of all reads) of archaeal-related sequences belonging to Thaumarchaeota, Nanoarchaeota, Thermoplasmata and Bathyarchaeia (see Table S2). Thus, it can be stated that none of methanogens, capable of converting DMS was presented in the initial pack ice samples.
- The text in Conclusions is a bit vague. In the first part of the section, the authors indicate the correspondence of the results with the already available data and explain the need for the study. In the second part, the authors describe the results in general terms. The article could benefit if all the results described in the text would be more clearly formulated, the most revealing data could be indicated and the main contribution of the work to the problem under study would be well defined.
ïƒ We took this remark into account, although it is worth to mention that all the results were already discussed in details in the Results and Discussion Section. However, we have added a paragraph in Conclusions (lines 683-693 in the modified version of manuscript), pointing readers to the main contribution of the work. Namely, the appearance in the stimulated fast ice of bacterial groups that are atypical for the sea ice-associated microbiota found in pack ice in the light summer season.
Reviewer 3 Report
The paper is very interesting. It can be accepted with minor modifications.
1- The Introduction appears to long. In my opinion it can be reduced.
2- the text in figure 2 is difficult to read. The font should be incresed.
3- the discussion lacks a comparison with the microbial communities from other studies conducted in Antarctica, just to better appreciate the results of this study. For example, it would be interesting to compared this study with the results obtained in 10.1007/s00248-015-0568-9.
Author Response
First of all, we would like to express our gratitude for positive evaluation of our manuscript.
Responses to the Reviewer#3.
- The Introduction appears to long. In my opinion it can be reduced.
ïƒ Following this comment, part of the Introduction is shortened by deleting a paragraph (lines 76-90 in the old version of manuscript) that is not directly related to the present work.
2- the text in figure 2 is difficult to read. The font should be increased.
ïƒ We made corrections regarding this Figure accordingly.
3- the discussion lacks a comparison with the microbial communities from other studies conducted in Antarctica, just to better appreciate the results of this study. For example, it would be interesting to compared this study with the results obtained in 10.1007/s00248-015-0568-9.
ïƒ It is not entirely true. On the lines 437-440 in the modified version of the manuscript we wrote: “The phylum Proteobacteria (represented by the classes Alpha- and Gammaproteobacteria) and the phylum Bacteriodetes were the dominant taxa, as previously reported for microbial communities associated with Antarctic first-year ice and multiyear ice [15,21,22,27,47-54].
So, we had in mind and referred to 11 articles describing microbial communities associated with Antarctic sea ice. However, we have cited recommended article of Pucciarelli et al, 2015 [54] in the modified version of manuscript, as suggested.